# Multiplex profiling of 16 immune checkpoints identifies novel serum biomarker panels for breast cancer detection and TNBC stratification: A case-control study

Mouna Stayoussef[1], Azza Habel[1], Weili Xu[2], Mariem Bessaad[1], Hanen Bouaziz[3], Mouna Ayadi [3], Wassim Y. Almawi [1,4]*, Anis Larbi[2,5], Besma Yacoubi-Loueslati[1]

**1** University of Tunis El Manar (UTM), Faculty of Sciences of Tunis (FST), Laboratory of Mycology, Pathologies and Biomarkers (LR16ES05), Tunis, Tunisia, **2** Singapore Immunology Network (SIgN), Agency for Science Technology and Research (A* STAR), Immunos Building, Singapore, Singapore, **3** Salah Azaiez Oncology Institute, Bab Saadoun, Tunis, Tunisia, **4** Department of Biological Sciences, Brock University, St. Catharines, Ontario, Canada, **5** Beckman Coulter Life Sciences, Villepinte, France

* wassim.almawi@fst.utm.tn

## Abstract

### Background

Immune checkpoints (ICs) are key regulators of anti-tumor immunity, yet their diagnostic potential as blood-based biomarkers in breast cancer (BC) remains insufficiently characterized. Comprehensive serum profiling using multiplex immunoassays may enable minimally invasive detection and molecular stratification, particularly for triple-negative breast cancer (TNBC).

### Methods

Serum samples from 88 treatment-naïve BC patients and 49 age-matched controls were analyzed using a 16-analyte MILLIPLEX® immuno-oncology panel. Six co-inhibitory and ten co-stimulatory IC proteins were quantified. Diagnostic accuracy was assessed using ROC curves and logistic regression. Associations with TNBC subtype, chemotherapy response, and 6-month progression-free survival (PFS) were evaluated.

### Results

Seven immune checkpoint proteins (LAG-3, BTLA, CD80, GITRL, CTLA-4, GITR, TLR-2) showed significant differential expression between BC patients and controls. A seven-protein panel demonstrated high diagnostic accuracy (AUC = 0.89; sensitivity 83%; specificity 86%), surpassing CA15−3 and CEA. TNBC patients exhibited a distinct eight-protein signature, with TIM-3, CTLA-4, and CD28 independently

**Data availability statement:** The datasets generated and/or analyzed during the current study are available in the Mendeley repository (Almawi, Wassim (2026), "Almawi_16-IC_Multiplex", Mendeley Data, V1) and can be accessed via the following DOI: https://doi.org/10.17632/p57yrjz9ds.1.

**Funding:** The author(s) received no specific funding for this work.

**Competing interests:** The authors have declared that no competing interests exist.

associated with TNBC classification. Elevated baseline TIM-3 and PD-L1 were associated with chemotherapy resistance and shorter PFS.

## Conclusions

Comprehensive serum IC profiling identified biomarkers with strong diagnostic and subtype-discriminatory potential. These minimally invasive panels show potential for BC detection and TNBC stratification, pending validation in prospective and longitudinal studies. Validation in larger, multi-center cohorts is warranted.

---

## 1. Introduction

Breast cancer (BC) remains the leading cancer diagnosis among women globally, with over 2.3 million new cases and 670,000 deaths annually [1,2]. Tumor recognition depends on immune surveillance balancing activation and inhibition, with immune checkpoints (ICs) crucial for self-tolerance and anti-tumor regulation [3,4]. In this regard, it was shown that dysregulated ICs promote immune evasion and tumor progression. Despite improvements in screening and treatment, early detection remains critical for reducing mortality. Current serum biomarkers, including CEA and CA15–3, demonstrate limited sensitivity and specificity (with corresponding 95% CIs) for early-stage disease. In contrast, prognostic markers such as ER, PR, HER2, and Ki-67 provide subtype classification but do not reliably predict treatment response [5,6]. This diagnostic gap underscores the urgent need for novel, minimally invasive biomarkers that enable earlier detection and more precise risk stratification.

Immune checkpoints (ICs) are increasingly studied as diagnostic and prognostic biomarkers in breast cancer (BC) [7,8]. Key IC molecules include PD-1/PD-L1 [9,10], CTLA-4, and LAG-3 [4,11], which regulate tumor immunity by suppressing T-cell activity [3,12]. This suppression promotes tumor survival, metastasis, and immune tolerance [8,13]. ICIs were developed to restore antitumor responses and have shown success in melanoma, lung cancer, and other malignancies [10,13,14]. However, their role in BC remains unclear. Most studies assess IC expression in tumour tissue, which is invasive and heterogeneous [15]. On the other hand, soluble ICs in circulation may offer a more practical measure of systemic immune dysregulation [16,17], but serum-based profiling is still limited.

The strong association of ICs with altered immune activity and tumour aggressiveness prompted interest in their potential diagnostic and prognostic roles in BC risk stratification and treatment decisions [18]. Several IC mediators were reported, including BTLA, CD27, CD28, CD40, CD80, CD86, CTLA-4, GITR, GITRL, HVEM, ICOS, LAG-3, PD-1, PD-L1, TIM-3, and TLR-2 [7,8]. Widely expressed in tumors, PD-L1 promotes tumor progression, and blockade of PD-1/PD-L1 interaction with atezolizumab was reportedly efficacious in metastatic TNBC [9,19]. On the other hand, the efficacy of CTLA-4 inhibition via ipilimumab is limited by resistance, prompting the development of combinatorial ICI strategies [14,20]. Further research is needed to clarify the biomarker and therapeutic potential of ICs in BC [11]. Prior

studies focused on tissue-based IC expression, mainly PD-1, PD-L1, and CTLA-4 [21], with limited data on systemic serum profiles. No multiplex serum analysis of 16 ICs has been reported, and their value for TNBC stratification remains largely unexplored.

To date, no study has undertaken a comprehensive serum-based quantification of 16 co-inhibitory and co-stimulatory immune checkpoint proteins, nor has it evaluated their collective diagnostic accuracy or biological relevance for TNBC classification. We hypothesize that systemic immune checkpoint deregulation is detectable in circulation and can be used to enhance early breast cancer detection and subtype differentiation [4,21]. This represents a novel and impactful approach that addresses a critical gap in systemic immune profiling for breast cancer. Our assay measured six co-inhibitory (BTLA, TIM-3, LAG-3, CTLA-4, PD-1, PD-L1) and ten co-stimulatory (CD27, CD28, CD40, HVEM, TLR-2, GITR, GITRL, ICOS, CD80, CD86) proteins, offering a comprehensive immune profile. This minimally invasive method supports early detection and personalized treatment through serum-based biomarker panels.

## 2. Patients and methods

### 2.1 Patients

This case-control study enrolled 88 women with histopathologically confirmed breast cancer (BC), recruited from the surgical and oncology departments of the Salah Azaiez Institute (SAI) in Tunis between March 2019 and November 2020. The study was approved by the Ethics Committee of SAI, with approval code ISA/2018/19, granted on June 25, 2018. All participants were treatment-naïve at the time of inclusion. The study adhered to the Reporting Recommendations for Tumor Marker Prognostic Studies (REMARK) checklist [22]. Inclusion criteria for BC patients were histologically confirmed invasive BC, 18 years of age or older, no prior cancer treatment including chemotherapy, radiotherapy, or hormonal therapy, and ECOG performance status 0–2 [1,23]. Exclusion criteria included concurrent malignancies, active autoimmune diseases or immunosuppressive conditions, current use of immunosuppressive medication, active infections, and pregnancy or lactation. The cancer-free control group consisted of 49 healthy, age-matched (±5 years) women with no personal or first-degree family history of cancer or immune disorders and were selected to establish baseline differences in immune checkpoint profiles. However, the inclusion of disease controls was beyond the scope of this discovery-phase study (addressed in future validation work). Control subjects were required to have no personal or family history of cancer and no autoimmune disorders.

Clinical and pathological data were collected through structured interviews and medical record review, including tumor stage, grade, hormone receptor status (estrogen receptor [ER], progesterone receptor [PR], and human epidermal growth factor receptor 2 [HER2]), and planned treatment regimens. Patients received standardized institutional regimens built around anthracycline- and taxane-based combinations. In triple-negative disease, neoadjuvant AC-T or TC (docetaxel–cyclophosphamide) was commonly used, with platinum agents such as carboplatin added in high-risk or BRCA-associated cases. After initiating first-line chemotherapy, patients were monitored for six months to evaluate treatment response. Sample size was based on published effect sizes for serum biomarkers in BC. To detect a moderate-to-large effect (Cohen's d = 0.65) with 80% power, ≥ 40 subjects per group were required; our sample (88 BC, 49 controls) provided 85% power for d ≥ 0.6. The TNBC subgroup analysis (n = 23 vs. 65) had 78% power to detect significant effects (d ≥ 0.80), but limited power (0.52) to detect moderate effects. ROC simulations support our sample's precision for AUC estimation (±0.05), meeting diagnostic accuracy guidelines.

All patients and controls provided written informed consent before their inclusion in the study. The research was conducted in accordance with the ethical standards of the 1964 Declaration of Helsinki. Clinical data abstraction was performed by investigators blinded to IC protein levels. All experimental procedures were conducted in accordance with the SAI guidelines for the care and use of laboratory subjects. The study was approved by the Ethics Committee of SAI, with approval code ISA/2018/19, granted on June 25, 2018.

## 2.2 Blood collection

To minimize pre-analytical variability, including Circadian and dietary influences on immune markers, all samples were collected in a standardized morning fasting state (8:00–10:00 AM), processed within a fixed time window (2 hours of venipuncture), and handled using uniform centrifugation (2,500 rpm for 20 minutes) and storage protocols (−80°C pending further analysis). Samples from BC patients were obtained before any cancer treatment (surgery, chemotherapy, radiotherapy, or hormonal therapy), ensuring that biomarker measurements were not influenced by treatment exposure. Venous blood samples (5 mL) were collected from patients and control women in sterile plain tubes (no preservatives). The resulting supernatant was transferred to a clean microcentrifuge tube and centrifuged for an additional 10 minutes at 14,000 rpm to remove cell debris and fragments. Hemolysis index and lipemia index were measured to exclude compromised specimens. Hemolyzed (n = 2) and lipemic (n = 1) samples were excluded, all of which were restricted to a single freeze–thaw cycle; samples were analyzed in randomized batches to limit systematic bias. These measures were implemented to ensure comparability of analyte measurements across study groups.

## 2.3 Multiplex immunoassay

To enable high-resolution profiling of immune dysregulation in breast cancer, serum concentrations of BTLA, CD27, CD28, CD40, CD80/B7-1, CD86/B7-2, CTLA-4, GITR, GITRL, HVEM, ICOS, LAG-3, PD-1, PD-L1, TIM-3, and TLR-2 IC proteins were measured using the MILLIPLEX MAP® Human Immuno-Oncology Checkpoint Protein Magnetic Bead Panel (Millipore Sigma, Burlington, MA). Laboratory personnel performing multiplex assays were blinded to patient diagnosis and clinical characteristics. Samples were coded with unique identifiers, and the code was not broken until after all assay results were finalized. Serum samples were thawed on ice, gently vortexed, centrifuged at 10,000 × g for 10 minutes at 4°C to remove particulates, diluted 1:2 in assay buffer, and analyzed in duplicate. This multiplex platform, combined with the Bio-Plex® 200 System and Bio-Plex Manager 5.0 software, enabled accurate, high-throughput detection across a wide dynamic range spanning three logs (0.14-12.5 pg/mL to 500–5000 pg/mL, depending on analyte).

Samples were randomized across plates to reduce batch effects, processed under blinded conditions, and analyzed in duplicate, with averaged values reported. Standard curves ($R^2 > 0.99$) were generated using five-parameter logistic regression, and stringent quality control required intra-assay CV < 15%, inter-assay CV < 20%, recovery of 85–115% for spiked samples, and detection limits of 0.14–12.5 pg/mL; samples exceeding CV > 20% were retested or excluded. Validation confirmed linearity ($R^2 > 0.99$), accuracy (92–108% recovery), precision (CV 4.2–16.8%), sensitivity (LOD 0.14–12.5 pg/mL), and specificity (<5% cross-analyte interference). Inter-assay variability was monitored with pooled controls (CV 7.3–16.8%), and assay drift was tracked using Levey–Jennings plots, ensuring accurate quantification of serum IC levels. Additional quality-control procedures were applied, which included duplicate measurements, curve-fit checks, and evaluation of outliers, influence, and batch effects, alongside sensitivity analyses such as log-transformation and exclusion of extreme values. This confirmed signal consistency and revealed no assay drift or batch-specific bias.

## 2.4 Statistical analysis

All statistical analyses were conducted using SPSS v29.0 (IBM, Armonk, NY), R version 4.2.0 (R Foundation for Statistical Computing, Vienna, Austria), and GraphPad Prism version 7.0 (GraphPad Software, San Diego, CA); 2-sided $p$-values < .05 were considered significant unless otherwise noted. Given the exploratory nature and interdependence of IC pathways, we report both unadjusted p-values and Bonferroni-corrected thresholds, and interpret findings based on effect size and biological plausibility. Data distribution was assessed using Shapiro-Wilk tests and Q-Q plots, revealing non-normality in 14 of 16 IC proteins (p < .05), prompting nonparametric analyses. Continuous variables were summarized as medians with interquartile ranges (IQR), and categorical variables as frequencies with percentages. Inter-group comparisons were done using Mann-Whitney U tests (two groups) or Kruskal-Wallis with Dunn's post-hoc (multiple groups), and chi-square or Fisher's exact tests for categorical data with low counts.

Effect sizes were calculated using Cohen's d for continuous and Cramér's V for categorical variables, interpreted as small (d = 0.2, V = 0.1), moderate (d = 0.5, V = 0.3), or large (d = 0.8, V = 0.5). Spearman's rank correlation (ρ) assessed monotonic relationships among IC proteins, visualized via corrplot in R, with correlation strength classified as weak (ρ < 0.3), moderate (0.3 ≤ ρ < 0.6), or strong (ρ ≥ 0.6). ROC curve analysis was used to evaluate diagnostic performance, with AUCs and 95% confidence intervals (CIs) calculated using DeLong's method. Optimal cut-offs were determined via Youden's index. Corresponding diagnostic performance metrics, including sensitivity, specificity, positive predictive value (PPV), negative predictive value (NPV), and likelihood ratios, were calculated with 95% CIs using standard binomial methods. Sensitivity analyses (with corresponding 95% CIs) excluding mild outliers (>3 × IQR) were performed; results remained unchanged. Logistic regression identified independent predictors of BC and TNBC, with model fit assessed using the Hosmer-Lemeshow test and multicollinearity assessed using VIFs (<5). Although formal internal validation procedures, including bootstrapping or cross-validation, were not performed due to the exploratory nature of the study, they will be addressed in future studies that incorporate internal validation methods (e.g., bootstrap resampling, k-fold cross-validation) alongside external validation. Cox regression was used to examine IC levels and 6-month PFS, with assumptions verified using Schoenfeld residuals. Kaplan-Meier curves and log-rank tests were used to compare survival. Missing data (<2%) were MCAR (*p* = .62), allowing complete-case analysis. All methods adhered to the biomarker and STARD guidelines.

## 3. Results

### 3.1 Study subjects

Table 1 summarizes demographic and clinical characteristics of the 88 women with BC and 49 healthy controls. The BC cohort consisted of 88 treatment-naïve women (mean age 53 ± 11 years, BMI 29.0 ± 5.2 kg/m²) with histopathologically confirmed disease. FIGO staging showed 38.6% early-stage, 40.9% advanced-stage, and 54.6% with distant metastases; TNBC cases represented 26.1% (n = 23) of the cohort. Clinical data included tumor stage, grade, receptor status (ER, PR, HER2), and treatment (chemotherapy, surgery, radiation). All BC patients received six cycles of Taxol-carboplatin chemotherapy in combination with carboplatin, pre- or post-surgery, based on tumor characteristics. The control group (n = 49) was well matched with cases for age (p = 0.32), BMI (p = 0.24), menopausal status (p = 0.58), smoking status (p = 0.74), and ethnicity (North African/Tunisian). No controls had active infections or were on immunosuppressants at enrollment, supporting the validity of the comparisons.

### 3.2 Differential expression of serum immune checkpoints in BC

Comparative analysis revealed significant differential expression of seven IC proteins between BC patients and controls, each demonstrating moderate-to-large effect sizes (Fig 1 and Table 2). Five proteins were significantly downregulated in BC. These comprised LAG-3 (median 6,008 vs. 93,688 pg/mL in controls; p < 0.001), which demonstrated the largest absolute difference between groups; this marked separation was consistent across samples and was supported by a large effect size (d = 0.89). Marked differences were also seen for BTLA (d = 0.67, p = 0.002), CD80 (d = 0.65, p = 0.003), GITRL (d = 0.63, p = 0.003), and CTLA-4 (d = 0.42, p = 0.04). On the other hand, GITR (d = 0.52, p = 0.01) and TLR-2 (d = 0.48, p = 0.01) were significantly upregulated. After Bonferroni correction (α = 0.003), LAG-3, BTLA, CD80, and GITRL remained statistically significant. The remaining nine ICs showed no significant differences between groups (S1 Table).

We next evaluated the diagnostic performance of these seven biomarkers individually and as a combined classifier. ROC curve analysis evaluated the clinical relevance of the seven differentially expressed ICs (Fig 2). To facilitate clinical implementation, we established optimal diagnostic cut-off values for individual IC proteins and the combined panel using Youden's index. For the seven-protein panel, the optimal cut-off yielded a PPV of 84% and an NPV of 85% in this cohort. At this threshold, the panel correctly classified 87% of all samples (95% CI: 80.2-91.5%). Individual protein cut-offs

**Table 1. Demographic and clinical characteristics of breast cancer patients and healthy controls.**

| Variable | BC Patients (N = 88) | Controls (N = 49) | *p*-value |
|---|---|---|---|
| **Age (years)**[1] | 53 ± 11 | 56 ± 13 | .32 |
| **BMI (kg/m²)**[1] | 29.0 ± 5.2 | 27.7 ± 4.5 | .24 |
| **Menopausal status** (Postmenopausal)[2] | 64% (56/88) | 59% (29/49) | .58 |
| **Smoking status** (Current/former)[2] | 12% (11/88) | 10% (5/49) | .74 |
| Distant metastasis[2] | 48 (54.6) | – | NA |
| **FIGO staging**[2] | | | |
| **Early stage** | 34 (38.6) | – | NA |
| **Late-stage** | 36 (40.9) | – | NA |
| **Unknown** | 18 (20.5) | – | NA |
| **Grade**[2] | | | |
| **Low-grade** | 43 (48.9) | – | NA |
| **High grade** | 28 (31.8) | – | NA |
| **Unknown** | 17 (19.3) | – | NA |
| **Estrogen receptor**[2] | | | |
| **Positive** | 66 (75.0) | – | NA |
| **Unknown** | 3 (3.4) | – | NA |
| **Progesterone receptor**[2] | | | |
| **Positive** | 63 (71.6) | – | NA |
| **Unknown** | 5 (5.7) | – | NA |

[1]Mean ± SD.

[2]Number (percent total).

were LAG-3 < 156 pg/mL (sensitivity = 78%, specificity = 82%), BTLA <89 pg/mL (sensitivity = 71%, specificity = 73%), and CD80 < 234 pg/mL (sensitivity = 69%, specificity = 74%).

LAG-3 exhibited the strongest individual performance (AUC [95% CI] = 0.82 [0.74–0.89]), followed by BTLA (AUC [95% CI] = 0.74 [0.64–0.85]) and CD80 (AUC [95% CI] = 0.73 [0.62–0.84]), while the remaining four ICs showed moderate accuracy (AUC range: 0.62–0.68). A combined seven-protein panel, developed using logistic regression, demonstrated strong diagnostic performance compared with healthy controls (AUC [95% CI] = 0.89 [0.83–0.94]), with high sensitivity (83%) and specificity (86%) at the optimal cut-off. This exceeded the performance of conventional BC markers such as CA15–3 (AUC [95% CI] = 0.72 [0.59–0.82]) and CEA (AUC [95% CI] = 0.68 [0.57–0.81]) in a subset of patients (n = 45; Fig 2C). The seven-protein panel demonstrated significantly higher AUC than CA15–3 (ΔAUC = 0.17, p = 0.002) and CEA (ΔAUC = 0.21, p < 0.001), with net reclassification improvement (NRI) of 32% (95% CI: 18–46%, p < 0.001). Sensitivity analyses showed that neither excluding extreme values nor transforming skewed variables materially altered the overall results, including the contribution of LAG-3 to model performance.

Correlation network analysis revealed significant interactions among IC proteins and their ligands, highlighting coordinated expression patterns in both BC patients and controls (Fig 3). Strong receptor-ligand correlations included CTLA-4 with CD80 (ρ = 0.45, p < 0.001), GITR with GITRL (ρ = 0.52, p < 0.001), and PD-1 with PD-L1 (ρ = 0.38, p = 0.002). Co-inhibitory receptors also showed notable inter-correlations: CTLA-4 with LAG-3 (ρ = 0.41), PD-1 with TIM-3 (ρ = 0.36), and BTLA with LAG-3 (ρ = 0.33), suggesting convergent signaling or shared regulatory mechanisms. Among co-stimulatory molecules, CD28 correlated with CD40 (ρ = 0.39) and ICOS (ρ = 0.35) (Fig 3).

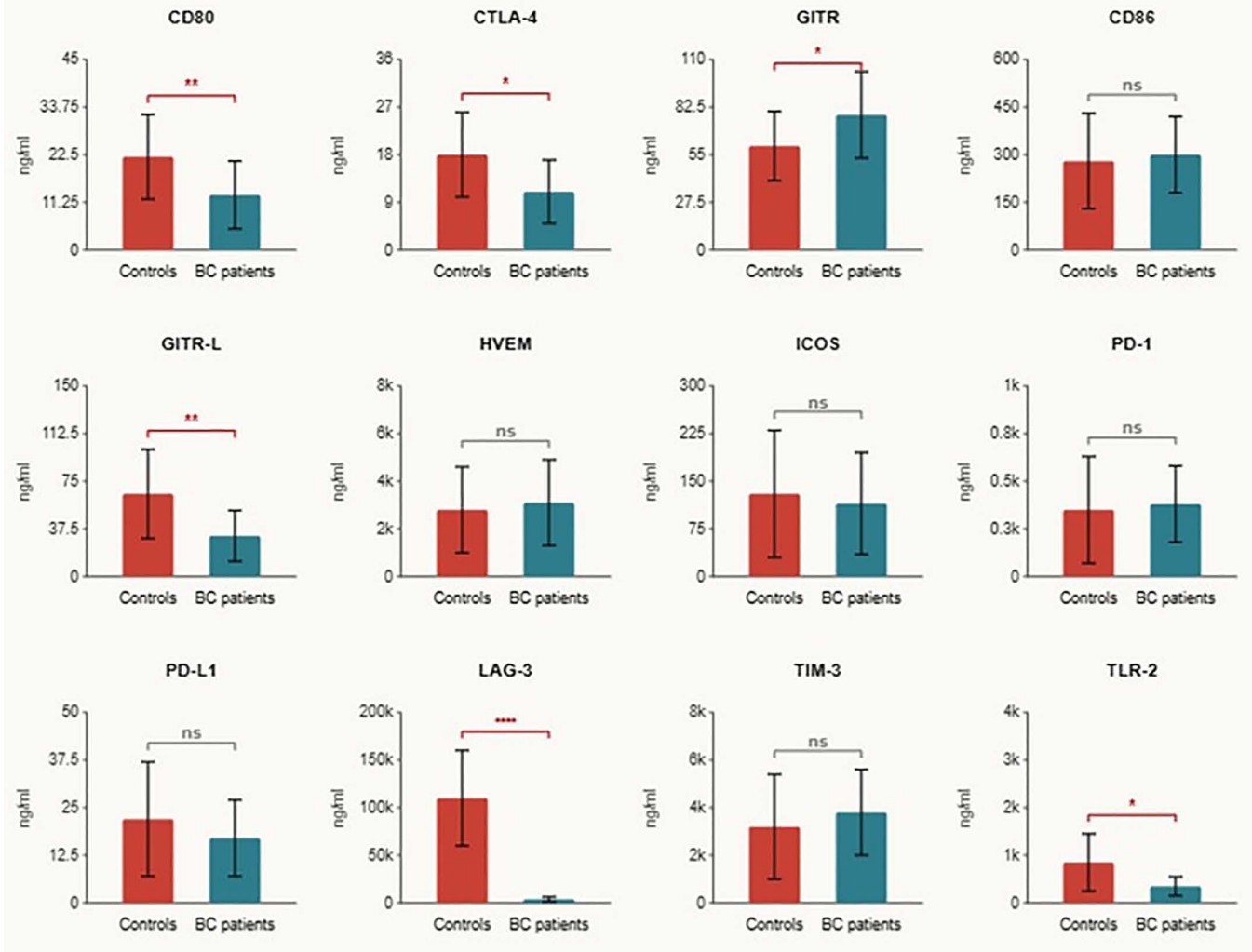

**Fig 1. Serum immune checkpoint levels in BC patients versus healthy controls.** Bar graphs display the mean concentrations (± standard error) of CD80, CTLA-4, IDO, CD86, GITRL, HVEM, ICOS, PD-1, PD-L1, LAG-3, TIM-3, and TLR-2 IC proteins. Statistically significant differences between groups are indicated as * $p < .05$, **$p < .005$, ***$p < .0001$.

### 3.3 Associated analytes with features of BC

TNBC patients (n = 23, 26.1%) exhibited a distinct IC signature compared to non-TNBC cases (n = 65; Table 3). This TNBC-related signature included both co-stimulatory (CD27, CD28, CD40, TLR-2) and co-inhibitory (CTLA-4, HVEM, LAG-3, TIM-3) molecules, highlighting the immunological complexity of this aggressive subtype. All eight proteins showed moderate effect sizes (Cohen's d: 0.51-0.67), indicating clinically meaningful differences. Principal component analysis (PCA) of these eight proteins successfully separated TNBC from non-TNBC with 78% accuracy. We identified independent predictors of BC using binary logistic regression, including the differentially expressed immune checkpoints BTLA, CD80, CTLA-4, GITR, GITRL, LAG-3, and TLR-2, as well as age, BMI, and menopausal status as clinical covariates.

Multicollinearity was assessed (all VIFs < 3.2) and found acceptable. After adjusting for confounders, the final parsimonious model, selected via backward stepwise elimination, retained three variables independently associated with BC in

**Table 2. Serum immune checkpoint levels in BC patients and healthy controls.**

| IC Protein | Controls: Median (IQR) pg/mL | BC Patients: Median (IQR) pg/mL | p-value[1] | Adjusted p-value[2] | B-H FDR[3] | Cohen's d (95% CI)[3] | Direction[4] |
|---|---|---|---|---|---|---|---|
| LAG-3 | 93,688 (61,450−126,200) | 6,008 (3,200−10,500) | <0.001 | <0.001 | 0.005 | 0.89 (0.57-1.21) | ↓ |
| BTLA | 77.2 (45.3-112.0) | 40.2 (22.1-65.8) | 0.002 | 0.032 | 0.005 | 0.67 (0.36-0.98) | ↓ |
| CD80 | 14.9 (8.2-22.5) | 8.7 (5.1-14.2) | 0.003 | 0.048 | 0.005 | 0.65 (0.34-0.96) | ↓ |
| GITRL | 58.1 (35.2-85.6) | 18.5 (10.2-32.8) | 0.003 | 0.048 | 0.005 | 0.63 (0.32-0.94) | ↓ |
| GITR | 36.2 (22.5-54.8) | 58.4 (38.2-78.5) | 0.010 | 0.160 | 0.012 | 0.52 (0.21-0.83) | ↑ |
| TLR-2 | 472.4 (285.0-688.5) | 365.2 (245.0-515.8) | 0.010 | 0.160 | 0.012 | 0.48 (0.17-0.79) | ↓ |
| CTLA-4 | 9.5 (5.8-14.2) | 8.1 (4.9-11.8) | 0.040 | 0.640 | 0.040 | 0.42 (0.11-0.73) | ↓ |

**B-H FDR**: Benjamini–Hochberg False Discovery Rate; **BTLA**: Band T lymphocyte attenuator; **CD**: Cluster of differentiation; **CTLA-4**: cytotoxic T-lymphocyte–associated antigen 4; **GITR**: glucocorticoid-induced; **GITRL**: glucocorticoid-induced ligand; **HVEM**: herpes virus entry mediator; **ICOS**: Inducible costimulator; **LAG-3**: Lymphocyte-Activation Gene 3; **PD-1**: programmed death-1; **PD-L1**: programmed death-ligand 1; **TIM-3**: T cell immunoglobulin mucin-3; **TLR-2**:Toll like receptor 2

[1]p-values calculated using Mann-Whitney U test.

[2]Adjusted p-value after Bonferroni correction for 16 comparisons (α = 0.003).

[3]FDR q-values were calculated using the Benjamini–Hochberg method

[4]Effect sizes calculated as Cohen's d with 95% confidence intervals.

[5]↓ indicates downregulation in BC patients; ↑ indicates upregulation in BC patients

this cohort: LAG-3 (OR = 0.23, p = .002), BTLA (OR = 0.31, p = .008), and TLR-2 (OR = 3.45, p = .005). Comparing TNBC and non-TNBC, multivariate analysis revealed that elevated TIM-3 (OR = 4.12, p = 0.002), CTLA-4 (OR = 3.78, p = .004), and CD28 (OR = 2.91, p = .015) were independently associated with the TNBC subtype, although these findings should be interpreted cautiously given the limited TNBC sample size (n = 23), and lack of model validation which increases the risk of overfitting. Although not validated, this TNBC association model achieved a c-statistic of 0.79 (95% CI: 0.69-0.88) in the study cohort. Hosmer-Lemeshow p-value of 0.58 and correctly classified 76% of TNBC cases with 81% specificity. As optimal cut-offs are data-driven and cohort-specific, their associated performance metrics should be interpreted with consideration of sampling variability. Complete associations with age, BMI, stage, grade, ER status, and metastasis are detailed in S2 Table.

Of the 72 patients who received taxane-platinum chemotherapy and had an evaluable response at 6 months, 18 (25%) were classified as non-responders (SD/PD), and the remaining 54 (75%) were responders (CR/PR). Compared with responders, baseline TIM-3 (p = 0.021) and PD-L1 (p = 0.035) levels were significantly higher in non-responders. On the other hand, higher baseline GITR levels were associated with a favorable treatment response (p = 0.018). Furthermore, PR-positive patients exhibited reduced levels of PD-1 (p = 0.04) (Table 3). These findings warrant validation in larger cohorts with longer follow-up.

During the 6-month follow-up period, 14 patients (15.9%) experienced disease progression despite chemotherapy. Baseline levels of TIM-3 (HR [95% CI] = 2.87 [1.32-6.24], p = 0.008) and PD-L1 (HR [95% CI] = 2.31 [1.11-4.79], p = 0.025) were significantly associated with shorter progression-free survival (PFS) in univariate Cox regression. In multivariate analysis adjusting for stage, grade, and molecular subtype, TIM-3 remained independently associated with shorter PFS (adjusted HR [95% CI] = 2.45 [1.08-5.56], p = 0.032), although this finding should be considered preliminary given the limited number of progression events (n = 14) that restricts the robustness of this observation. While Kaplan-Meier analysis showed that patients with high TIM-3 levels had shorter 6-month PFS than those with low TIM-3 levels (4.2 vs. 5.8 months; log-rank p = 0.006), this was based on short-term follow-up and requires confirmation in longer-term studies.

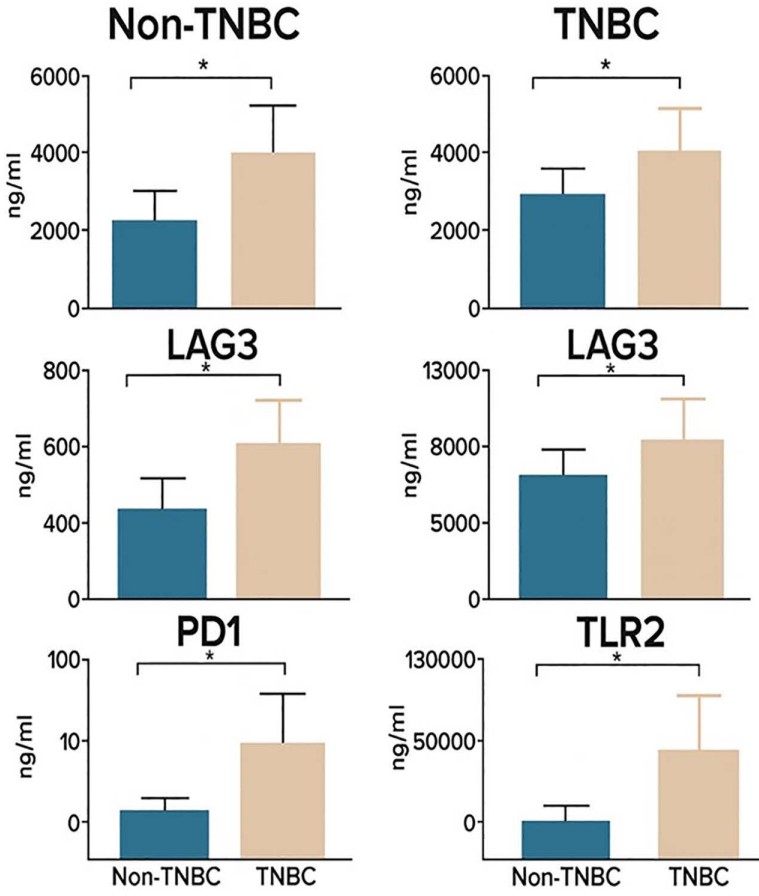

**Fig 2. Differential expression of IC proteins in TNBC versus non-TNBC patients.** Mean serum concentrations (± standard error) of CD28, HVEM, CD40, LAG-3, PD-1, TIM-3, CTLA-4, and TLR-2 IC proteins in TNBC (TN(+), pink bars) and non-TNBC (TN(–), blue bars) patients. Significant differences in expression levels between groups are indicated by red asterisks (*).

## 4. Discussion

This discovery-driven multiplex evaluation quantified 16 serum immune checkpoint proteins to diagnose BC and stratify TNBC. A seven-protein panel (LAG-3, BTLA, CD80, GITRL, CTLA-4, GITR, TLR) achieved high diagnostic accuracy, outperforming conventional BC markers (CA15−3, CEA) in this case–control framework. An eight-protein TNBC signature was identified, with TIM-3, CTLA-4, and CD28 as markers associated with TNBC and clinical outcomes in this cohort. Baseline TIM-3 and PD-L1 levels were associated with chemotherapy resistance and short-term progression-free survival; TIM-3 showed a potential association with short-term progression, pending validation in studies with longer follow-up. Unlike tissue-based assays, which are limited by invasiveness and heterogeneity [20,21], this serum-based 16-plex approach offers a minimally invasive, systems-level readout of immune dysregulation. Serum IC profiling appears promising for BC diagnosis and molecular subtyping, though its clinical decision-making utility requires further validation [24,25]. Because these data derive from patients with confirmed BC, they reflect diagnostic discrimination rather than true early detection, and performance must be assessed in pre-diagnostic or population-based cohorts. Without evaluation in larger, independent, or diverse samples, findings remain in the discovery phase. Beyond diagnostic relevance, these patterns may also inform BC immunobiology, as discussed below.

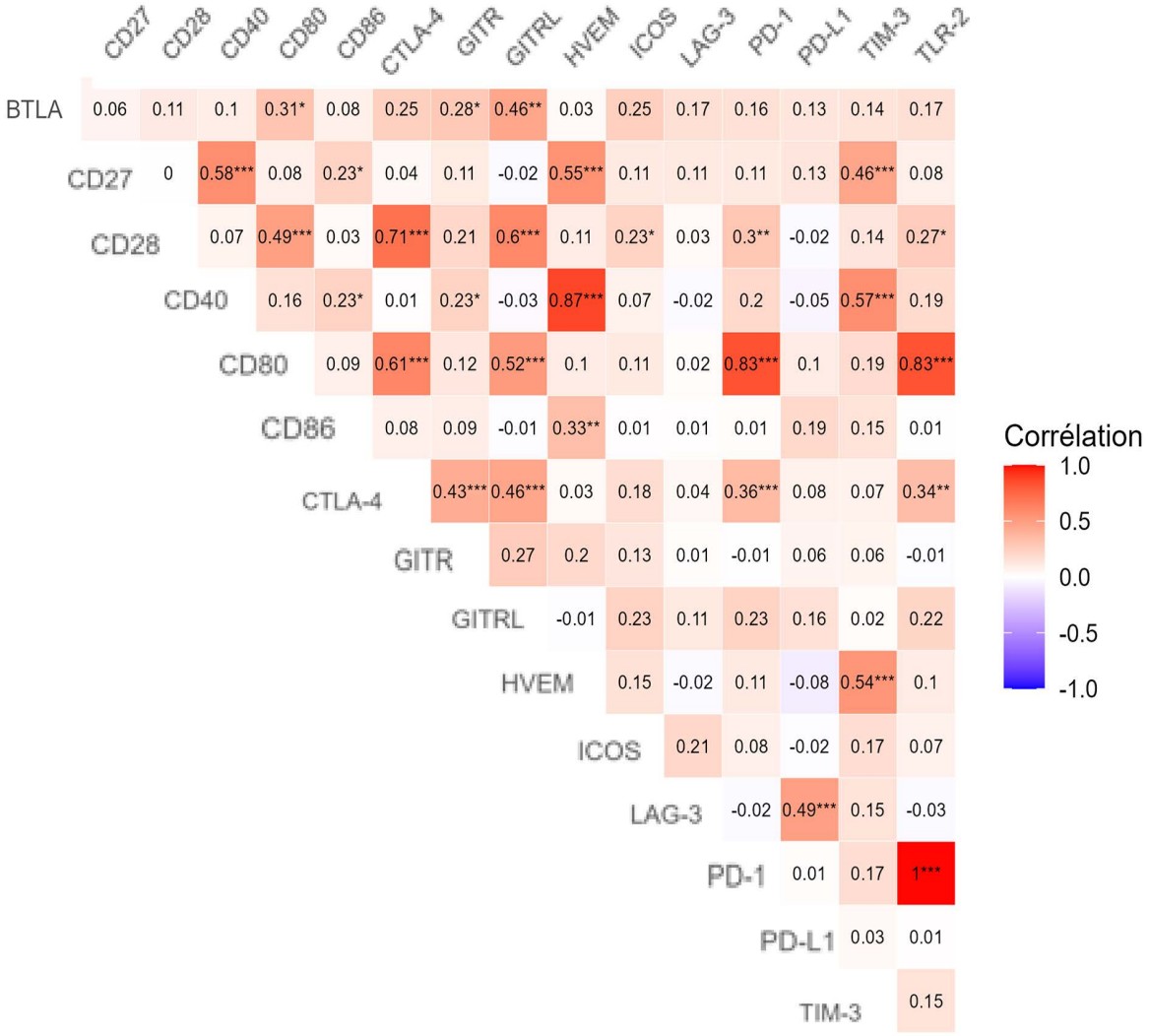

**Fig 3. Spearman rank correlation heatmap of serum IC proteins in breast cancer patients.** Heatmap illustrates pairwise Spearman correlation coefficients among BTLA, CD27, CD28, CD40, CD80, CD86, CTLA-4, GITR, GITRL, HVEM, ICOS, LAG-3, PD-1, PD-L1, TIM-3, and TLR-2 IC proteins. Color intensity reflects correlation strength and direction, with blue indicating positive correlations and red indicating negative correlations.

The coherent directional pattern of circulating immune checkpoint alterations likely reflects broader immune dysregulation in BC [4,7]. Several molecules involved in co-inhibitory signaling and antigen-presenting cell–T-cell interaction, LAG-3, BTLA, CD80, and GITRL, were reduced, a profile consistent with altered exhaustion dynamics or redistribution of checkpoint activity to the tumor microenvironment [8,13] and with their established roles in regulating T-cell activation and tolerance [9,11,13]. In contrast, the relative increase in GITR may represent compensatory activation or subset-specific regulation, given its association with enhanced T-cell proliferation and Treg modulation [26]. This combined pattern suggests complex remodeling rather than simple suppression or activation, aligning with evidence that soluble checkpoints capture systemic immune perturbations in malignancy, although their relationship to tumor-infiltrating immunity remains unresolved [12,26]. These interpretations remain provisional, underscoring the need for studies that integrate circulating, cellular, and tissue-level data to clarify their biological significance.

**Table 3. Serum immune checkpoint profiles associated with TNBC and clinical outcomes.**

| IC Protein | Non-TNBC[1] (n = 65) | TNBC[1] (n = 23) | p-value[2] | Effect Size (Cohen's d) | Clinical Relevance |
|---|---|---|---|---|---|
| CD28 | 752 ± 558 | 3,002 ± 4,735 | 0.01 | 0.67 | TNBC predictor |
| CD40 | 846 ± 541 | 1,325 ± 916 | 0.02 | 0.58 | TNBC predictor |
| CD27 | 2,693 ± 2,290 | 4,930 ± 3,707 | 0.10 | 0.51 | TNBC enriched |
| CTLA-4 | 9.3 ± 5.5 | 22.7 ± 19.8 | 0.03 | 0.54 | TNBC predictor |
| HVEM | 3,952 ± 2,790 | 4,974 ± 1,403 | 0.03 | 0.52 | TNBC enriched |
| LAG-3 | 7,894 ± 7,785 | 14,626 ± 1,390 | 0.02 | 0.59 | TNBC enriched |
| TIM-3 | 3,546 ± 2,056 | 4,920 ± 1,716 | 0.02 | 0.61 | TNBC predictor |
| TLR-2 | 431 ± 250 | 3,729 ± 8,733 | 0.02 | 0.55 | TNBC enriched |
| | **Responders (n = 54)** | **Non-responders (n = 18)** | | | |
| TIM-3 | 3,324 ± 1,890 | 4,458 ± 2,215 | 0.021 | 0.54 | Resistance marker[4] |
| PD-L1 | 18.2 ± 11.5 | 24.6 ± 14.8 | 0.035 | 0.48 | Resistance marker[4] |
| GITR | 68.5 ± 38.2 | 48.3 ± 22.1 | 0.018 | 0.52 | Response marker |
| | **PR-positive (n = 63)** | **PR-negative (n = 20)** | | | |
| PD-1 | 244 ± 152 | 1,039 ± 3,285 | 0.04 | 0.43 | Hormone receptor |

Abbreviations: **CTLA-4**, cytotoxic T-lymphocyte-associated antigen 4; **GITR**, glucocorticoid-induced TNFR-related protein; **HVEM**, herpes virus entry mediator; **LAG-3**, lymphocyte activation gene-3; **PD-1**, programmed death-1; **PD-L1**, programmed death ligand-1; **PR**, progesterone receptor; **SD**, standard deviation; **TIM-3**, T-cell immunoglobulin mucin-3; **TLR-2**, Toll-like receptor 2; **TNBC**, triple-negative breast cancer.

[1] Mean ± SD.

[2] p-values calculated using Mann-Whitney U test.

[3] Effect sizes calculated as Cohen's d. Only immune checkpoints with significant associations ($p < 0.05$) or clear trends ($p < 0.10$) are shown.

[4] Independent predictors in multivariate analysis adjusting for stage, grade, and Ki-67.

Correlations between ICs and their ligands revealed coordinated regulatory networks linked to clinical features and treatment response [27]. These findings support the concept of systemic IC dysregulation in BC, which is detectable in serum and amenable to multi-analyte, minimally invasive diagnostics to guide personalized therapy. While tumor biopsies remain essential, they are limited by invasiveness, cost, and heterogeneity in advanced disease [21,28]. For population-level early detection, biomarkers must be sensitive, reproducible, and compatible with low-cost routine testing; however, current multiplex platforms remain limited by cost and complexity. Thus, these immune checkpoint panels should be viewed as a discovery framework to develop simplified, cost-effective assays (e.g., ELISA or point-of-care) for scalable clinical implementation [29]. BC cells exploit IC pathways to evade immunity [29]; our data show upregulation of GITR and TLR-2, and downregulation of BTLA, CD80, CTLA-4, GITRL, and LAG-3, indicating immune suppression. Notable correlations between CTLA-4/CD80 and GITR/GITRL suggest actionable regulatory axes [30]. Insofar as treatment-related factors may influence immune checkpoint dynamics and should be considered when interpreting circulating biomarker profiles.

Previous studies reported reduced CD80 and CTLA-4 expression in BC patients, especially in early-stage cases [21,31]. In contrast, our study found elevated CTLA-4 and PD-1 levels in triple-negative BC (TNBC), aligning with prior reports of increased CTLA-4 in BC sera [32]. CTLA-4 binding to CD80 suppresses T-cell proliferation and survival [33], and soluble CTLA-4 (sCTLA-4) acts as an immunosuppressive molecule, reducing CD4 + /CD8 + T-cell proliferation and cytokine secretion (IFN-γ, IL-17A, IL-10) [34]. sCTLA-4 also competes with CD28 for binding to CD80/CD86, thereby inhibiting co-stimulatory signalling [35]. CD28 was upregulated in our TNBC cohort, which also showed elevated CD28 +, CTLA-4 +, and PD-1 + T cells, as well as double-positive CD28 + CD4 +, PD-1 + CD4 +, and PD-1 + CD8 + T cells, consistent with findings from a recent Brazilian study [36,37], highlighting immune dysregulation in TNBC.

Murine TNBC models demonstrated that dual CTLA-4 and PD-1 blockade, with adoptive cell therapy, delays tumor progression and prolongs survival by depleting regulatory T cells (Tregs) [38]. This, in turn, enhances antitumor immunity by reshaping lymphocyte and myeloid populations [39]. BTLA, a T cell-bound inhibitory receptor, is downregulated in BC and suppresses ERK1/2 signaling via HVEM interaction, thereby limiting tumor growth [40]. However, TNBC patients exhibit elevated HVEM and soluble HVEM levels, which impair naïve T cell activation and infiltration, thereby promoting immune evasion [41]. These findings highlight the therapeutic promise of targeting multiple immune checkpoints and their ligands to counteract immunosuppression in aggressive BC subtypes [3,7,18].

Similar to CTLA-4, LAG-3 is a co-inhibitory receptor involved in antitumour immune regulation and is downregulated in BC patients. In a Chinese study, LAG-3 was identified as a TNBC biomarker that inhibits T-cell activation and cytokine release while promoting Treg-mediated immunosuppression [42]. TIM-3, often co-expressed with PD-1, suppresses effector T-cell function, and its upregulation of PD-1, TIM-3, and LAG-3 in TNBC supports combination immunotherapy strategies [43,44]. Furthermore, GITR was elevated, while its ligand, GITRL, was reduced in BC [44], promoting immunosuppression via CD40, CD54, EpCAM downregulation and TGF-β elevation [45]. Lower GITR levels in metastatic BC were associated with a better prognosis, whereas higher levels were associated with improved survival across cancers [46]. The marked differences, notably for LAG-3, warrant caution, as platform-specific effects or true biological variability contributing to the wide dynamic range cannot be fully excluded, despite extensive quality control and sensitivity analyses.

Serum-based IC panels offer key advantages over tissue-based assays, including minimal invasiveness and the ability to enable systemic immune profiling, with potential for longitudinal assessment in future studies [7,42]. Our seven-protein panel achieved 89% diagnostic accuracy, outperforming conventional BC markers such as CA15−3 and CEA, particularly in early-stage disease. In TNBC, distinct IC signatures (CD27, CD28, CD40, CTLA-4, HVEM, LAG-3, TIM-3, TLR-2) reflect an immunosuppressive microenvironment driven by elevated co-inhibitory checkpoints, supporting therapeutic targeting [21]. Upregulation of CTLA-4, LAG-3, TIM-3, and PD-1 provides a rationale for combination checkpoint blockade, with ongoing trials testing anti-PD-1/PD-L1 agents in combination with anti-CTLA-4 or anti-LAG-3 [47]. Baseline TIM-3 and PD-L1 levels also correlate with chemotherapy resistance, suggesting their potential to guide upfront immunotherapy or combination regimens, pending prospective validation [16,47].

CD40 was significantly elevated in TNBC patients, suggesting its role in immune escape by suppressing specific T-cell subsets [48], and correlating with aggressive disease features [49]. Similarly, high serum levels of soluble TLR-2 (sTLR-2) were detected, indicating its potential as a biomarker for breast cancer susceptibility and TNBC aggressiveness [50]. Elevated TLR-2 levels are associated with poorer outcomes, especially in the mesenchymal-like TNBC subtype, consistent with an Egyptian study that identified sTLR-2 as a candidate biomarker [17]. High TLR-2 expression in TNBC is associated with poor overall survival in TLR-2-positive mesenchymal-like cases [51,52]. These findings, supported by elevated TLR-2 levels in our TNBC cohort, underscore the roles of CD40 and TLR-2 in tumor progression and their potential clinical utility in risk stratification and therapeutic targeting. Although strict pre-analytical procedures and assay quality controls were implemented, the significant difference in LAG-3 concentrations between cases and controls warrants cautious interpretation. Residual variability due to sample handling, matrix effects, or platform-specific behaviour cannot be completely ruled out, highlighting the need for independent validation using alternative assays and external cohorts.

Our study has several notable strengths, in particular the comprehensive multiplex approach that simultaneously quantifies 16 IC proteins with 89% accuracy and 85% NPV, outperforming conventional markers, and providing an unprecedented systems-level view of immune dysregulation in BC. The blinded sample processing and rigorous assay validation enhanced the validity of the results. In addition, the identification of two distinct IC panels (a seven-protein diagnostic and an eight-protein TNBC panel) demonstrates the specificity and biological relevance of our findings. Furthermore, the multiplex assay requires only 50 μL of serum, takes 4–6 hours to complete, and costs $200–300 per sample.

Despite these advantages, our study has limitations that warrant consideration. A major limitation is its cross-sectional design, as all immune checkpoint measurements were taken at a single baseline time point, thus precluding assessment

of IC dynamics. The associations identified in this study should not be interpreted as predictive, prognostic, or suitable for treatment monitoring, underscoring the need for longitudinal designs to determine whether these markers reflect evolving immune states or track clinical outcomes. Generalizability remains limited by the single-centre cohort, modest sample size, and underpowered subgroups—particularly TNBC and progression events, which increase the likelihood of unstable estimates and render subtype- or outcome-specific signals exploratory. Interpretation is further constrained by the exclusive use of healthy controls, uncertainty regarding the cellular origins of soluble checkpoints, and the absence of data from asymptomatic or diagnostically ambiguous populations, while the multiplex platform may not yet meet feasibility thresholds for broader screening. The 6-month follow-up restricts evaluation of meaningful progression and survival endpoints, and the regression strategy lacked internal resampling or external validation, increasing susceptibility to overfitting. Given the cross-sectional, baseline-only design, temporal or causal inferences cannot be drawn. All associations, therefore, remain exploratory and require prospective validation, particularly for analytes with extreme or skewed distributions.

## 5. Conclusion

This study offers the first comprehensive serum-based profiling of 16 immune checkpoint proteins in breast cancer, identifying two panels with promising potential for stratification, subject to confirmation in prospective and longitudinal cohorts. Although limited by its cross-sectional design, moderate sample size, and short follow-up, the research provides a foundation for biomarker development while highlighting the need for studies capable of predicting, monitoring treatment, and assessing long-term outcomes. The regression models also require internal resampling and external validation to reduce overfitting and establish their true predictive value. Broader multicentre cohorts with greater ethnic diversity and suitable disease-control groups will be crucial for evaluating reproducibility, clinical specificity, and real-world diagnostic performance. The absence of internal validation limits confidence in model stability and highlights the need for validation in future studies that focus on independent replication, pre-diagnostic assessment, assay standardization, and integration with established clinical markers, while also addressing the practical challenge of translating these findings into cost-effective platforms for routine laboratory use and personalized breast cancer management.

## Supporting information

**S1 Table. Complete serum immune checkpoint protein levels in breast cancer patients and healthy controls.**
(PDF)

**S2 Table. Complete comparative analysis of serum immune checkpoint levels across clinical and demographic subgroups in breast cancer patients.**
(PDF)

## Acknowledgments

We extend our gratitude to the medical and laboratory staff at Salah Azeiz Hospital for their invaluable assistance throughout this study.

   **Consent to participate:** Informed consent was obtained from all individual participants included in the study.

## Author contributions

**Conceptualization:** Mouna Stayoussef, Wassim Y. Almawi.

**Data curation:** Mouna Stayoussef, Azza Habel, Mariem Bessaad, Hanen Bouaziz.

**Formal analysis:** Mouna Stayoussef, Weili Xu, Mouna Ayadi.

**Investigation:** Anis Larbi, Besma Yacoubi-Loueslati.

**Methodology:** Azza Habel, Weili Xu, Mariem Bessaad.

**Project administration:** Besma Yacoubi-Loueslati.

**Supervision:** Wassim Y. Almawi.

**Writing – original draft:** Mouna Stayoussef, Azza Habel, Wassim Y. Almawi.

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
