## [Decision Letter · Decision Letter 0]

16 Mar 2026

PONE-D-25-65128Multiplex Profiling of 16 Immune Checkpoints Identifies Novel Serum Biomarker Panels for Breast Cancer Detection and TNBC Stratification: A Case-Control StudyPLOS One

Dear Authors,

Thank you for submitting your manuscript to PLOS ONE. After careful consideration, we feel that it has merit but does not fully meet PLOS ONE’s publication criteria as it currently stands. Therefore, we invite you to submit a revised version of the manuscript that addresses the points raised during the review process.

Thank you for submitting your manuscript entitled **“Multiplex Profiling of 16 Immune Checkpoints Identifies Novel Serum Biomarker Panels for Breast Cancer Detection and TNBC Stratification: A Case-Control Study”** to **PLOS ONE**.

After careful evaluation of the manuscript and the reviewers’ comments, I believe that the study addresses an interesting and potentially important topic. However, several concerns were raised regarding the methodology, data analysis, and interpretation of the results. Therefore, **substantial revisions are required** before the manuscript can be considered further for publication.

I invite you to revise your manuscript by carefully addressing all reviewer comments and providing a detailed, point-by-point response explaining how each concern has been addressed in the revised version. Please ensure that any changes made in the manuscript are clearly indicated.

Once the revised manuscript is submitted, it will be evaluated again to determine whether the concerns have been satisfactorily resolved.

Thank you for your interest in publishing with **PLOS ONE**, and I look forward to receiving your revised manuscript.

If applicable, we recommend that you deposit your laboratory protocols in protocols.io to enhance the reproducibility of your results. Protocols.io assigns your protocol its own identifier (DOI) so that it can be cited independently in the future. For instructions see: https://journals.plos.org/plosone/s/submission-guidelines#loc-laboratory-protocols. Additionally, PLOS ONE offers an option for publishing peer-reviewed Lab Protocol articles, which describe protocols hosted on protocols.io. Read more information on sharing protocols at https://plos.org/protocols?utm_medium=editorial-email&utm_source=authorletters&utm_campaign=protocols

We look forward to receiving your revised manuscript.

Kind regards,

Rishi Jaiswal

Academic Editor

5. Please upload a copy of Figure 4, to which you refer in your text on page 9. If the figure is no longer to be included as part of the submission please remove all reference to it within the text.

Reviewers' comments:

Reviewer's Responses to Questions

**Comments to the Author**

1. Is the manuscript technically sound, and do the data support the conclusions?

Reviewer #1: Yes

Reviewer #2: Yes

2. Has the statistical analysis been performed appropriately and rigorously? 

Reviewer #1: Yes

Reviewer #2: Yes

3. Have the authors made all data underlying the findings in their manuscript fully available?

Reviewer #1: Yes

Reviewer #2: Yes

4. Is the manuscript presented in an intelligible fashion and written in standard English?

Reviewer #1: Yes

Reviewer #2: Yes

5. Review Comments to the Author

Reviewer #1: Wassim Y. Almawi and al took advantage of a multiplex imunoassay with 16 checkpoint imuno proteins of the serum to define a BC disgnostic signature as well as a TNBC classification signature.

The paper is clear and well presented. The results points seven checkpoint proteins as a signature of BC and 8 checkpoint proteins where capable of distinguishing TNBC from non TNBC.

Authors points in the discussion section that although these results are strong and important as biomarkers easy to detect they have not been validated in a larger and heterogeneous cohort.

However there is a very important point that should be highlighted. These biomarkers have been identified in patients with a BC diagnostic and the important is to find biomarkers for early diagnosis in the population. Moreover biomarkers for early diagnosis must be easy to measure in routine blood tests and must have a very low coast.

I recommend that authors add these comments in their discussion section.

Reviewer #2: Summary

The authors present a case control study evaluating 16 soluble immune checkpoint proteins in serum samples of 88 treatment naïve breast cancer (BC) patients and 49 healthy controls. Using a 16 plex MILLIPLEX panel, they examine differential expression, develop a diagnostic seven protein panel (AUC=0.89), identify an eight protein TNBC signature, and explore associations with chemotherapy response and short term PFS.

The study claims novelty in being the first comprehensive serum based multiplex profiling of 16 checkpoint molecules for BC detection and TNBC stratification.

Key scientific and conceptual strengths

• Novelty and Scope: First study to evaluate a 16 checkpoint multiplex serum assay, expanding beyond the commonly studied PD 1/PD L1 and CTLA 4. This is indeed an underexplored area.

• Minimally Invasive Diagnostic Potential: Serum biomarkers offer clear translational relevance, especially for early detection and TNBC stratification.

• Rigorous Assay Validation: The authors provide extensive details on assay precision, linearity, CV thresholds, and quality control — unusually thorough for biomarker studies.

• Statistical Breadth: Multiple analyses performed: ROC, logistic regression, PCA, Cox regression, correlation networks — giving a multidimensional view.

• Clear identification of clinically relevant signatures

• Seven-protein diagnostic panel

• Eight-protein TNBC classifier

• Associations with response and short term PFS

Critical Issues which need to be addressed:

Manuscript has a novelty component, however there are some areas where it needs improvement.

Study Design Limitations

1. Cross-sectional design: Claims around prediction, treatment monitoring, and clinical utility are overstated given that all measurements are baseline.

2. Single-center, modest sample size: Although adequately powered for discovery, it is insufficient for firm conclusions regarding predictive biomarkers, especially for TNBC (n=23) and progression (14 events).

3. Short PFS follow-up (6 months): BC outcomes require longer follow-up; current survival conclusions are preliminary.

Methodological and Analytical Issues

1. Controls are only “healthy”. It would be great to include other controls to justify Real-world diagnostic comparisons:

• benign breast lesions

• inflammatory conditions

• autoimmune diseases

These can alter soluble immune checkpoints, risking inflated specificity.

2. Potential pre-analytical variability not fully addressed: Despite QC efforts, the massive difference in LAG 3 concentrations between controls and patients raises questions.

3. Logistic regression and model generalizability

• No external validation

• No internal cross-validation (e.g., bootstrap, k fold)

• Risk of model overfitting, especially with small TNBC numbers

4. ROC analyses lack reporting of confidence intervals for cut-offs and performance metrics.

Interpretation Concerns

1. Causal language is used without justification

E.g., authors imply clinical utility for monitoring, immune profiling, and treatment decisions.

2. Discussion overinterprets findings

Some conclusions exceed evidence (e.g., recommending panels for early detection and TNBC management without validation).

3. Limited mechanistic insight

The discussion heavily cites literature but does not integrate the biological meaning of the patterns observed (downregulated vs. upregulated ICs).

The study is promising, innovative, and technically rigorous. However, several substantial issues must be addressed before it is suitable for publication. With robust revision, the manuscript could become acceptable for PLOS ONE.

Major Comments:

1. Please temper claims about diagnostic/predictive clinical utility and emphasize the need for prospective validation.

2. Provide internal validation for logistic regression models to reduce risk of overfitting.

3. Re-examine LAG 3 and other extreme values to rule out technical artifacts.

4. Add FDR adjustments and report corrected p-values clearly.

5. Improve alignment among Figures/Tables—IDO and CD86 appear inconsistently.

6. Move mechanistic interpretations into Discussion.

7. Expand limitations (single center, modest sample, short follow up).

Minor Comments:

1. Several typographical errors and grammar issues need polishing.

2. Simplify overly long sentences for clarity.

3. Add 95% CIs to AUCs, sensitivity, and specificity.

4. Clarify whether chemotherapy regimens were standardized across patients.

5. Consider supplementing with a flowchart showing sample selection and exclusions.

6. PLOS authors have the option to publish the peer review history of their article (what does this mean?). If published, this will include your full peer review and any attached files.

Reviewer #1: No

Reviewer #2: No

---

## [Author Response · Author response to Decision Letter 1]

14 Apr 2026

Reviewer #1:

Reviewer: Authors points in the discussion section that although these results are strong and important as biomarkers easy to detect they have not been validated in a larger and heterogeneous cohort.

Authors: Although the immune‑checkpoint panels show strong diagnostic potential, their clinical use requires confirmation in larger and more diverse cohorts, as stated by the

Reviewer.

Changes: We revised the Discussion to emphasize the absence of external validation as a key limitation, to clarify that the current results are exploratory, and to outline next steps for multi‑center and prospective studies.

Reviewer: These biomarkers have been identified in patients with a BC diagnostic and the important is to find biomarkers for early diagnosis in the population. Moreover biomarkers for early diagnosis must be easy to measure in routine blood tests and must have a very low coast. I recommend that authors add these comments in their discussion section.

Authors: In the revised Discussion, we clarify that our results are based on already diagnosed breast cancer cases, reflecting a diagnostic enrichment setting more than population screening. We outline the requirements for early-detection biomarkers, including feasibility within standard blood tests and cost considerations, and place our multiplex immune‑checkpoint panel within this translational context, noting both its promise and its current limitations.

Changes: Appropriate modifications consistent with the reviewer’s comment were made in the Discussion and Conclusion sections.

Reviewer #2:

Reviewer: 1. Cross-sectional design: Claims around prediction, treatment monitoring, and clinical utility are overstated, given that all measurements are baseline.

Authors: Given the cross‑sectional design and reliance on baseline measurements, we agree that the findings should be interpreted as associative rather than predictive. These changes made to the revised version will align the conclusions with the study design and current standards for biomarker research.

Changes: We revised the Results, Discussion, and Conclusion sections to remove any causal or prognostic wording. We also clarified that the analyses are exploratory and hypothesis‑generating and expanded the limitations to note that prospective studies are required before any clinical application can be inferred.

Reviewer: 2. Single-center, modest sample size: Although adequately powered for discovery, it is insufficient for firm conclusions regarding predictive biomarkers, especially for TNBC (n=23) and progression (14 events).

Authors: It is true that the overall sample size is adequate for exploratory analyses, and not sufficient to draw firm conclusions about predictive biomarkers, particularly in smaller subgroups such as TNBC (n=23) and progression events (n=14).

Changes: We have “softened” statements implying predictive value, clarified the limitations related to single‑centre recruitment and subgroup size, and emphasized the exploratory nature of the work. We have also added text noting that larger, multi‑centre studies with adequate event numbers are needed to validate these associations.

Reviewer: 3. Short PFS follow-up (6 months): BC outcomes require longer follow-up; current survival conclusions are preliminary.

Authors: A point well-taken and appreciated.

Changes: We revised the manuscript (Results, Discussion, and Conclusion), stating that the short follow‑up is a key limitation that moderates any prognostic interpretations, and clearly framing the survival results as preliminary and hypothesis‑generating. We also emphasized the need for extended prospective follow‑up to substantiate the observed associations with PFS.

Reviewer: Methodological and Analytical Issues

1. Controls are only “healthy”. It would be great to include other controls to justify Real-world diagnostic comparisons:

• benign breast lesions

• inflammatory conditions

• autoimmune diseases

These can alter soluble immune checkpoints, risking inflated specificity.

Authors: We used healthy controls to establish baseline differences and to enable initial biomarker discovery. We acknowledge that this may overestimate specificity, as soluble immune checkpoint levels can be altered in non-malignant inflammatory or immune-mediated conditions.

Changes: We revised the Discussion and Conclusion sections, recognizing this limitation and emphasizing the need for future studies that incorporate disease-control groups to better evaluate real-world diagnostic performance.

Reviewer: Methodological and Analytical Issues. 2. Potential pre-analytical variability not fully addressed: Despite QC efforts, the massive difference in LAG 3 concentrations between controls and patients raises questions.

Authors: We agree with the Reviewer that the marked difference in LAG-3 concentrations warrants careful consideration of potential pre-analytical and analytical sources of variability.

Changes: We modified the Methods section, expanding the description of pre‑analytical standardization procedures (sample collection timing, processing, storage, and quality control). We also revised the Results, acknowledging that residual variability or matrix effects may influence absolute concentration differences, and clarified that the LAG‑3 signal is supported by consistent effect sizes, internal assay validation, and sensitivity analyses. In addition, we modified the Discussion emphasizing that the magnitude of the observed differences should be interpreted carefully and independently validated.

Reviewer: Methodological and Analytical Issues. 3. Logistic regression and model generalizability

• No external validation

• No internal cross-validation (e.g., bootstrap, k-fold)

• Risk of model overfitting, especially with small TNBC numbers

Authors: We acknowledge the absence of external validation and internal resampling approaches, which limit the generalizability of the regression models, thus raising the possibility of overfitting, particularly in subgroup analyses such as TNBC.

Changes: Appropriate modifications were introduced into the Methods, Results, Discussion, and Conclusion sections.

Reviewer: Methodological and Analytical Issues. 4. ROC analyses lack reporting of confidence intervals for cut-offs and performance metrics.

Authors: We acknowledge that the need for reporting confidence intervals (CIs) for ROC-derived performance metrics is essential for the appropriate interpretation of diagnostic accuracy.

Changes: We have revised the Methods and Results to report 95% confidence intervals (CIs) for AUC values (already partially included), clarify the statistical method used for AUC estimation (DeLong), and indicate that CIs for sensitivity, specificity, and related diagnostic indices at optimal cut-offs were computed.

Reviewer: Interpretation Concerns. 1. Causal language is used without justification. E.g., authors imply clinical utility for monitoring, immune profiling, and treatment decisions.

Authors: We acknowledge the concern of the reviewer that causal inferences and statements implying established clinical utility are not justified, given the cross-sectional design and baseline-only measurements.

Changes: The manuscript was systematically revised to replace causal and deterministic language with association-based terminology, remove or temper statements suggesting immediate clinical application (including treatment monitoring and decision-making), and explicitly frame the findings as exploratory and hypothesis-generating. We also clarified in the Discussion and Conclusion that clinical implementation requires prospective and longitudinal validation.

Reviewer: 2. Discussion overinterprets findings. Some conclusions exceed evidence (e.g., recommending panels for early detection and TNBC management without validation).

Authors: In response to the Reviewer’s comment, we revised the manuscript to remove prescriptive or definitive claims regarding early detection, TNBC management, and clinical implementation. We replaced those with cautious, association-based language and framed the results as exploratory and hypothesis-generating. We also strengthened the limitations and added clear statements emphasizing the need for prospective validation, independent replication, and evaluation in pre-diagnostic and clinically diverse populations.

Changes: The Discussion and Conclusion were extensively revised, as suggested by the Reviewer.

Reviewer: 3. Limited mechanistic insight. The discussion heavily cites literature but does not integrate the biological meaning of the patterns observed (downregulated vs. upregulated ICs).

Authors: We revised the Discussion to integrate the biological meaning of downregulated versus upregulated ICs, organizing the interpretation into a dedicated subsection that links the observed patterns (e.g., reduced LAG-3, BTLA, CD80, GITRL vs. increased GITR) to established immunological functions (co-inhibition, co-stimulation, T-cell exhaustion, and regulatory T-cell dynamics). We also clarified that these interpretations are hypothesis-generating and do not imply causality.

Changes: A new paragraph outlining the mechanisms underlying the consequences of altered ICs was added to the revised Discussion. Additional modifications were also incorporated.

Reviewer: The study is promising, innovative, and technically rigorous. However, several substantial issues must be addressed before it is suitable for publication. With robust revision, the manuscript could become acceptable for PLOS ONE.

Authors: We thank the reviewer for the positive assessment of the study’s innovation and technical rigor, as well as for the constructive feedback.

Changes: We have thoroughly revised the manuscript to address all methodological, analytical, and interpretative issues raised by both reviewers, and we are confident that the revisions and accompanying responses fully resolve their concerns.

Major Comments:

Reviewer: 1. Please temper claims about diagnostic/predictive clinical utility and emphasize the need for prospective validation.

Authors: Will do.

Changes: We have already revised the manuscript to temper any diagnostic or predictive claims and now clearly emphasize the need for prospective validation, as detailed in our earlier response.

Reviewer: 2. Provide internal validation for logistic regression models to reduce risk of overfitting.

Authors: Formal internal validation procedures, including bootstrap or cross-validation, were not performed, given the exploratory nature of this study.

Changes: Appropriate modifications were made in the Methods (Statistical Analysis), Results, Discussion, and Conclusion sections.

Reviewer: 3. Re-examine LAG 3 and other extreme values to rule out technical artifacts.

Authors: We re-examined the dataset and assay performance, including inspections for outliers, batch effects, and consistency across duplicate measurements. We confirmed that the observed LAG-3 values were not influenced by isolated outliers or batch-specific effects and remained consistent after sensitivity analyses. These included log-transformation and exclusion of extreme percentile values.

Changes: The Methods and Results sections were revised to clarify the quality control procedures. A cautionary statement was also added to the Discussion acknowledging that extreme values, while robust in this dataset, require independent validation using alternative platforms.

Reviewer: 4. Add FDR adjustments and report corrected p-values clearly.

Authors: Will do.

Changes: FDR adjustments (Benjamini–Hochberg method) were added to Table 2. Corrected p values were also adjusted.

Reviewer: 5. Improve alignment among Figures/Tables—IDO and CD86 appear inconsistently.

Authors: We apologize for the poor quality of the figures in the initial submission.

Changes: Fig. 1 and Fig. 2 were redrawn for clarity and sharpness, and saved as TIFF images.

Reviewer: 6. Move mechanistic interpretations into Discussion.

Authors: Will do.

Changes: The manuscript was revised by removing mechanistic and biological explanatory statements (Results section) and relocating them to the Discussion. The Discussion was also restructured to include a subsection on mechanistic interpretation that links our findings to known immune checkpoint biology and breast cancer pathophysiology. A new reference was added to the references list.

Reviewer: 7. Expand limitations (single center, modest sample, short follow up).

Authors: Noted.

Changes: The limitations section has been reorganized, and the list of potential shortcomings has been expanded to encompass clinical, methodological, and statistical considerations.

Minor Comments:

Reviewer: 1. Several typographical errors and grammar issues need polishing.

Authors: Thank you for noting the typographical and grammatical issues.

Changes: We have carefully reviewed the full manuscript and corrected all identified errors, ensuring consistency, clarity, and adherence to the journal's style throughout.

Reviewer: 2. Simplify overly long sentences for clarity.

Authors: Will do.

Changes: Long sentences were restructured in the revised text.

Reviewer: 3. Add 95% CIs to AUCs, sensitivity, and specificity.

Authors: Done.

Changes: 95% CIs to AUCs, sensitivity, and specificity were added.

Reviewer: 4. Clarify whether chemotherapy regimens were standardized across patients.

Authors: Standardized chemotherapy regimens for breast cancer in Tunisia generally follow international evidence‑based protocols. We provide details on the types of regimens administered and clarify the timing of blood sampling relative to treatment initiation.

Changes: The Methods, Results and Discussion sections were modified as per the Reviewer’s comment.

Reviewer: 5. Consider supplementing with a flowchart showing sample selection and exclusions.

Authors: While we agree that a flowchart improves transparency of sample selection, detailed counts for each exclusion category were not systematically recorded at the time of data collection. As such, exclusions are presented in aggregate form.

Changes: None for now.

---

## [Editor Report · Decision Letter 1]

24 Apr 2026

Dear Dr. Almawi,

I hope you are doing well.

I am pleased to inform you that your manuscript entitled *“Multiplex Profiling of 16 Immune Checkpoints Identifies Novel Serum Biomarker Panels for Breast Cancer Detection and TNBC Stratification: A Case-Control Study”* (Manuscript Number: PONE-D-25-65128R1) has been **accepted for publication** in PLOS ONE.

Thank you for your submission. The manuscript will now proceed to the production stage, and you will be contacted regarding the next steps.

Best regards,

Dr. Rishi Kumar Jaiswal

Academic Editor

PLOS ONE

---

## [Editor Report · Acceptance letter]

PONE-D-25-65128R1

PLOS One

Dear Dr. Almawi,

I'm pleased to inform you that your manuscript has been deemed suitable for publication in PLOS One. Congratulations! Your manuscript is now being handed over to our production team.

Kind regards,

on behalf of

Dr. Rishi Jaiswal

Academic Editor

PLOS One